# Cross-lingual Prompting: Improving Zero-shot Chain-of-Thought Reasoning across Languages

**Libo Qin**♣*, **Qiguang Chen**♠*, **Fuxuan Wei**♠, **Shijue Huang**◇, **Wanxiang Che**♠

♣ School of Computer Science and Engineering, Central South University, China
♠Research Center for Social Computing and Information Retrieval
♠Harbin Institute of Technology, China
◇Harbin Institute of Technology, Shenzhen, China
lbqin@csu.edu.cn, {qgchen, fxwei, car}@ir.hit.edu.cn
joehsj310@gmail.com

## Abstract

Chain-of-thought (CoT) is capable of eliciting models to explicitly generate reasoning paths, thus promoting reasoning accuracy and attracting increasing attention. Specifically, zero-shot CoT achieves remarkable improvements in a wide range of reasoning tasks by simply instructing the LLM with the prompt "Let's think step by step!". Despite the success of zero-shot CoT, the existing zero-shot prompting techniques remain limited to a single language, making it challenging to generalize to other languages and hindering global development. In this work, we introduce cross-lingual prompting (**CLP**), aiming to improve zero-shot CoT reasoning across languages. Specifically, CLP consists of two main components: (1) *cross-lingual alignment prompting* and (2) *task-specific solver prompting*. The cross-lingual alignment prompting is responsible for aligning representations across different languages, whereas the task-specific solver prompting is used to generate the final chain of thoughts and results for the reasoning task. In addition, we further introduce cross-lingual self-consistent prompting (**CLSP**) to ensemble different reasoning paths across languages. Our experimental evaluations on several benchmarks demonstrate that CLP and CLSP significantly outperform the existing prompting methods and achieve state-of-the-art performance. We hope this work will inspire further breakthroughs in cross-lingual CoT.

## 1 Introduction

Large Language Models (LLMs) have shown remarkable success across various NLP tasks (Qin et al., 2023; Hendy et al., 2023; Pan et al., 2023; Ziyu et al., 2023). Unlike the previous pre-trained language models (PLMs) (Devlin et al., 2019; He et al., 2021), LLMs are capable of achieving zero-shot learning without the need to modify the model parameters during the training and testing process,

---
*Equal Contribution

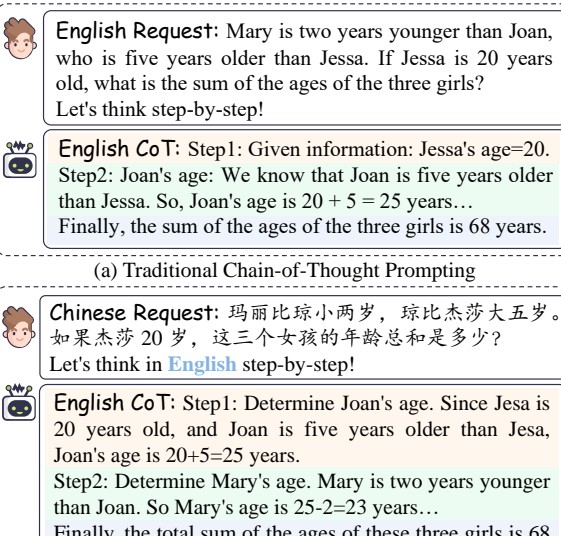

(a) Traditional Chain-of-Thought Prompting

(b) Cross-lingual Chain-of-Thought

Figure 1: Traditional Chain-of-Though (CoT) vs. Cross-lingual CoT.

which gains increasing attention. Specifically, zero-shot chain-of-thought (CoT) (Kojima et al., 2022) only needs to append the prompt "Let's think step by step!", which can elicit strong reasoning capabilities from large language models and demonstrate promising performance on various tasks, including arithmetic reasoning, commonsense reasoning (Wei et al., 2022; Kojima et al., 2022) and even robotic planning(Ahn et al., 2022; Huang et al., 2022). Take a traditional CoT in Figure 1 (a) as an example, a trigger prompt "Let's think step by step!" is provided along with an English request to perform step-by-step reasoning. Eventually, LLMs produce the corresponding answer "68 years".

In fact, there are over 200 countries and 7,000 languages worldwide. With the acceleration of globalization, there is an urgent need for generalizing the current CoT across different languages. Despite the remarkable success of zero-shot CoT, its reasoning abilities still struggle to generalize to different languages. Shi et al. (2022) introduce

the first multi-lingual dataset to evaluate the mathematical reasoning capabilities of language models to facilitate the research of cross-lingual CoT. Unlike traditional CoT scenarios, where the language of the request and CoT output is the same, cross-lingual CoT requires the LLM to generate CoT in English for any given language by providing a trigger sentence "Let's think in *English* step by step!", which is illustrated in Figure 1 (b). Unfortunately, little attention has been paid to zero-shot cross-lingual CoT.

To generalize the current CoT across languages, we propose a novel cross-lingual prompting (CLP), which aims to effectively bridge the gap across different languages. It consists of two components: (1) *Cross-lingual Alignment Prompting* and (2) *Task-specific Solver Prompting*. Specifically, the *cross-lingual alignment prompting* is used to align representations between different languages. In our experiments, instead of the traditional *"Let's think step by step"*, we use *"Let's understand the task in English step-by-step."*. The inherent intuition is that as model gradually understands the task in English, it inherently captures the relationship between the source language and English. After aligning the representations between different languages, we further utilize a *task-specific solve prompting* to complete the final task by setting *"Let's resolve the task you understand above step-by-step!"*. Such simple yet effective CLP can greatly enhance the reasoning ability of cross-lingual scenarios. Furthermore, inspired by the self-consistency work, we propose cross-lingual self-consistent prompting (CLSP), which enables the model to ensemble different views of reasoning paths across languages.

Experimental results reveal that CLP achieves the SOTA performance by outperforming all baselines with a gain of over 1.8%. In addition, CLSP can further enhance the performance by integrating knowledge across different languages. The main contributions of this work are concluded as follows:

- We introduce cross-lingual prompting that contains *cross-lingual alignment prompting* and *task-specific solver prompting*, which jointly improve zero-shot CoT reasoning across languages;

- We further propose cross-lingual self-consistent prompting to integrate reasoning paths across different languages;

- Extensive evaluations on several benchmarks

reveal that both CLP and CLSP are capable of improving zero-shot cross-lingual CoT effectively and achieving SOTA performance (with over 1.8% improvement on AVG accuracy).

We hope this work can inspire further research on cross-lingual CoT and the code are available at Cross-Lingual-Prompting.

## 2 Background

This section describes the definition of traditional and cross-lingual chain-of-thought.

### 2.1 Traditional Chain-of-Thought

Chan-of-thought is a powerful technique to elicit the strong reasoning ability of large language models (LLM), which is capable of completing complex tasks. For the traditional chain-of-thought (CoT) generation approach, LLM is appended as a simple prompt "Let's think step by step!" to output the specific reasoning paths, which is denoted as:

> Request: [Given sentence $X$]
> Let's think step by step!

### 2.2 Cross-lingual Chain-of-Thought

While traditional CoT has achieved remarkable success, it is limited to generating CoT within a single language and lacks effective cross-lingual transferability. Therefore, cross-lingual CoT aims to enable models to handle requests in any language and generate CoT specifically in the target language (i.e., English) (Shi et al., 2022).

## 3 Cross-lingual Prompting

To elicit the cross-lingual reasoning ability of LLM, we introduce cross-lingual prompting (CLP) as a solution. Specifically, CLP consists of two components: (1) *cross-lingual alignment prompting* (§3.1) and (2) *task-specific solver prompting* (§3.2).

### 3.1 Step 1: Cross-lingual Alignment Prompting

Cross-lingual alignment is a core challenge for cross-lingual transfer. Therefore, to better capture the alignment information, we first introduce cross-lingual alignment prompting (refer to Figure 2 (a)). Specifically, our approach initiates the LLM with a specific task of aligning information. The request is formulated as follows:

> Please act as an expert in multi-lingual understanding in [Source Language $L_s$].

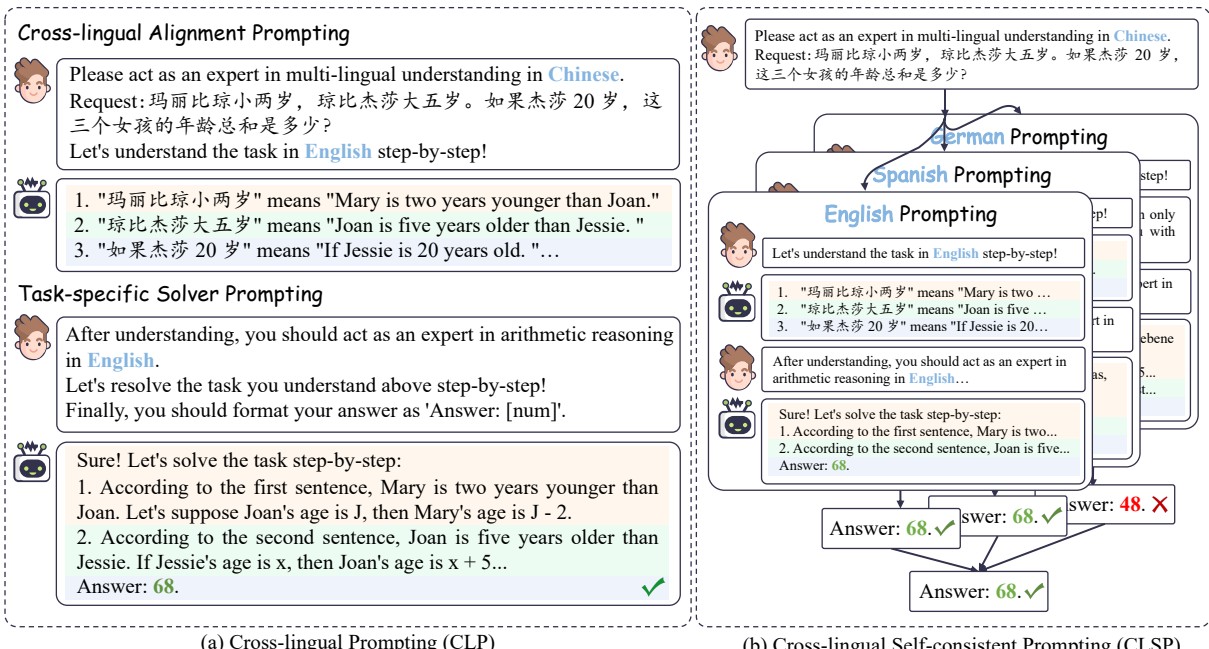

Figure 2: The main framework of CLP (a) and CLSP (b). CLP consists of *cross-lingual alignment prompting* and *task-specific solver prompting*, while cross-lingual self-consistent prompting (CLSP) aims to integrate various reasoning paths across different languages.

> Request: [Given sentence $X$]
> Let's understand the task in [Target Language $L_t$] step-by-step!

Given the sentence $X$, we first simulate the LLM's expertise in multi-lingual comprehension. Furthermore, we introduce a step-by-step alignment process from source language $L_s$ to target language $L_t$. The intermediate semantic alignments are represented as $\{a_i\}_{i=1}^{S}$, where $S$ denotes the number of alignment steps. Overall, the formulation of our *cross-lingual alignment prompting* method can be expressed as follows:

$$\mathcal{A} = \arg\max p(a_1, \ldots, a_S | X, L_s, L_t), \qquad (1)$$

where $\mathcal{A}$ denotes the alignment response in step 1.

### 3.2 Step 2: Task-specific Solver Prompting

After achieving cross-lingual alignment, we further propose *task-specific solver prompting* to facilitate multi-step reasoning in a multi-lingual setting.

Specifically, given the target language $L_t$, and the alignment text $\mathcal{A}$ obtained from the previous step, we prompt the LLM to engage resolving target tast $T$. And LLM tries to determine the final result $F_t$ along a multi-step reasoning path $R = \{r_i\}_{i=1}^{|R|}$, where $|R|$ represents the number of steps in the reasoning process, which is reg-

ulated by the LLM. Specifically, we design the *task-specific solver prompting* as:

> After understanding, you should act as an expert in [Target Task $T$] in [Target Language $L_t$].
> Let's resolve the task you understand above step-by-step!

Formally, the set of potential reasoning path $R$ is organized into the final reasoning path $\mathcal{R}_t$ for target language $L_t$, which can be determined as:

$$\mathcal{R}_t = \arg\max_R p(R | C, L_t, T), \qquad (2)$$

where $C$ represents the dialog history, including the input variables $X$, $L_s$, $L_t$, and $\mathcal{A}$.

Furthermore, we provide an instruction to format the model's answer, which is defined as:

> Finally, you should format your answer as 'Answer: [num]'.

Formally, the answer extraction is determined as:

$$F_t = \arg\max p(f | \mathcal{R}_t), \qquad (3)$$

where $F_t$ represents the text of the answer, generated from all potential reasoning result $f$.

## 4 Cross-lingual Self-consistent Prompting

In our research, we observe that LLMs show varying patterns across different languages. Inspired

| Model | bn | de | es | fr | ja | ru | sw | te | th | zh | AVG |
|---|---|---|---|---|---|---|---|---|---|---|---|
| GPT-3 (text-davinci-002) | | | | | | | | | | | |
| Direct (Shi et al., 2022) | 4.4 | 14.8 | 17.2 | 16.8 | 11.2 | 12.4 | 8.8 | 0.8 | 8.8 | 18.0 | 11.3 |
| Native-CoT[†] (Shi et al., 2022) | 6.4 | 36.0 | 40.4 | 37.6 | 26.0 | 28.4 | 11.2 | 0.4 | 10.8 | 40.0 | 23.7 |
| En-CoT[†] (Shi et al., 2022) | 9.6 | 44.0 | 44.8 | 46.0 | 32.4 | 28.4 | 20.8 | 5.6 | 19.6 | 40.8 | 29.2 |
| Translate-En[†] (Shi et al., 2022) | 41.2 | 46.4 | 51.6 | 46.4 | 44.8 | 48.8 | 37.6 | 42.8 | 41.2 | 47.2 | 44.8 |
| PaLM-540B | | | | | | | | | | | |
| Direct (Shi et al., 2022) | 17.2 | 18.8 | 20.0 | 19.6 | 16.0 | 22.0 | 15.6 | 17.6 | 16.8 | 19.2 | 18.3 |
| Native-CoT[†] (Shi et al., 2022) | 46.0 | 49.2 | 56.8 | 46.4 | 40.0 | 48.4 | 35.2 | 45.6 | 52.8 | 46.8 | 48.7 |
| En-CoT[†] (Shi et al., 2022) | 46.4 | 53.6 | 58.0 | 51.2 | 49.6 | 55.6 | 44.4 | 46.8 | 49.6 | 46.0 | 50.1 |
| Translate-En[†] (Shi et al., 2022) | 53.2 | 57.2 | 60.0 | 55.2 | 50.0 | 59.6 | 51.2 | 49.6 | 50.8 | 55.6 | 54.2 |
| GPT3.5 (gpt-3.5-turbo) | | | | | | | | | | | |
| Direct | 33.6 | 56.0 | 61.2 | 62.0 | 52.8 | 62.0 | 48.0 | 7.6 | 42.4 | 60.0 | 48.6 |
| Native-CoT | 26.4 | 70.0 | 70.4 | 64.4 | 52.8 | 62.4 | 54.0 | 10.4 | 40.0 | 59.6 | 51.0 |
| En-CoT | 50.0 | 73.6 | 69.6 | 70.0 | 60.4 | 65.6 | 55.2 | 22.0 | 48.0 | 63.2 | 57.8 |
| Translate-En | 66.4 | 75.6 | 74.4 | 72.4 | 66.0 | 72.8 | 69.6 | **58.0** | 57.6 | 71.6 | 68.4 |
| CLP | 64.8 | 80.0 | 82.4 | 79.2 | 69.2 | 81.6 | 74.8 | 38.8 | 62.0 | 73.6 | 70.6 |
| CLSP | **75.2** | **86.8** | 84.8 | 82.0 | **77.2** | **87.6** | **76.0** | 52.0 | **68.0** | **77.2** | **76.7** |

Table 1: Main results on MGSM. "`Direct`" denotes the original input request will be given to model directly. "`Native-CoT`" signifies that the model generates inference steps in the same language as the input. "`En-CoT`" indicates the given non-English input request and returned with English chain-of-thought result. "`Translate-En`" denotes we translate non-English input requests into English by Google translation API. [†] denotes the 6-shot results sourced from Shi et al. (2022).

by Wang et al. (2022), we propose a cross-lingual self-consistent prompting (CLSP) to integrate reasoning knowledge across different languages (as shown in Figure 2 (b)).

Specifically, for each step in the reasoning process, we require LLM to generate alignment responses in different target language $L_t$ and employ respective reasoning steps. Finally, we retain the answers that exhibit a high level of consistency in the inferred reasoning results ($f$) through a voting mechanism. These consistently inferred answers are then considered as the final result, which can be formulated as:

$$\hat{F} = \text{argmax} \sum_{t=1}^{|L|} \sum_{f}^{|f|} \mathbb{1}\left(F_t = f\right), \quad (4)$$

where $|L|$ represents the count of target languages, $|f|$ signifies the count of potential reasoning results $f$ across all target languages, and $\mathbb{1}(X)$ denotes a 0-1 function that returns 0 when $X$ is False and returns 1 when $X$ is True.

## 5 Experiments

### 5.1 Implementation Settings

We select three representative state-of-the-art pretrained large language models as baseline references for our study: GPT-3 (Brown et al., 2020),

PaLM (Chowdhery et al., 2022) and GPT3.5[1]. Following Wei et al. (2022) and Kojima et al. (2022), we evaluate the performance using accuracy score (Acc.). The top-p parameter in all processes is set to 1. We select the temperature in *Cross-lingual Alignment Prompting* from $[0, 2]$ and the temperature in *Task-specific Solver Prompting* from $[0, 1]$.

### 5.2 Main Results

The main results are illustrated in Table 1. From the results, we have the following observations:

**(1) GPT-3.5 exhibits notable cross-lingual reasoning superiority.** When evaluated in the all scenarios mentioned in Table 1, GPT-3.5 surpasses the few-shot results of PaLM-540B and GPT-3 by a significant margin (achieving improvements of 30.3%, 2.3% 7.7%, and 14.2% over PaLM-540B, respectively). As shown in Wang et al. (2023a), multi-lingual SFT and RLHF techniques lead to substantial improvement in cross-lingual reasoning performance.

**(2) CLP achieves state-of-the-art performance.** As depicted in Table 1, CLP surpasses all previous baselines, specifically outperforming PALM-540B(`Translate-En`) with an improvement of 16.4%. This improvement cannot be solely attributed to GPT-3.5 (CLP even achieves a 2.2%

[1] https://platform.openai.com/docs/guides/chat/introduction

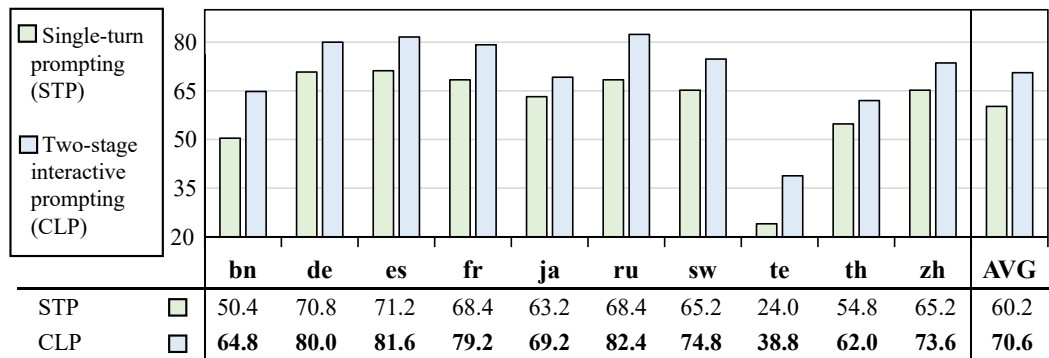

| | | bn | de | es | fr | ja | ru | sw | te | th | zh | AVG |
|---|---|---|---|---|---|---|---|---|---|---|---|---|
| STP | ☐ | 50.4 | 70.8 | 71.2 | 68.4 | 63.2 | 68.4 | 65.2 | 24.0 | 54.8 | 65.2 | 60.2 |
| CLP | ☐ | **64.8** | **80.0** | **81.6** | **79.2** | **69.2** | **82.4** | **74.8** | **38.8** | **62.0** | **73.6** | **70.6** |

Figure 3: The accuracy comparison between two-stage interactive prompting and single-turn prompting.

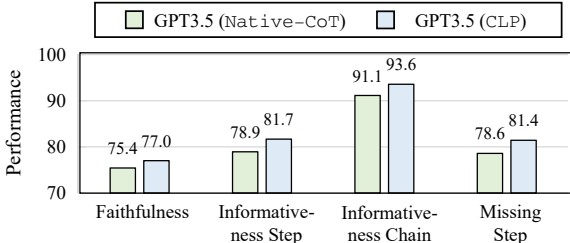

Figure 4: The analysis of Chain-of-Thought quality between GPT-3.5 (`Native-CoT`) and CLP.

higher average accuracy than `Translate-En`). These findings suggest that *cross-lingual alignment prompting*(CLP) goes beyond simple text translation and further enhances the model's inherent cross-lingual understanding capabilities.

**(3) CLSP further significantly improves performance.** As illustrated in Table 1, CLSP exhibits a remarkable superiority over CLP across all languages (with 6.1% improvements on average accuracy). This observation reveals that integrating knowledge across different languages can effectively boost the reasoning performance on cross-lingual CoT, verifying the effectiveness of cross-lingual self-consistent prompting.

## 5.3 CLP Analysis

### 5.3.1 CLP results better reasoning quality

To further investigate why CLP works, we employ the framework of ROSCOE (Golovneva et al., 2022) to evaluate the quality of the reasoning paths in the model's Chain of Thought. The implementation details are shown in Appendix A.2.

As shown in Figure 4, we find that the reasoning paths of CLP demonstrate higher faithfulness, exhibiting better consistency with key steps during the reasoning process. Specifically, the faithfulness score increased by 1.6%, indicating that the model

better understood the problem statement and ensured a clear inference chain without generating irrelevant or misused information. Furthermore, we observe 2.8% and 2.5% improvements in the Informativeness metrics for "Step" and "Chain", respectively. It suggests that the model's reasoning, after cross-lingual alignment, could provide more well-grounded inference steps. Additionally, CLP shows a 2.8% enhancement in the Miss-step metric, indicating that the model's reasoning could encompass a complete logical chain, leading to better performance.

### 5.3.2 Two-stage interactive prompting is better than single turn prompting

This section explores the effectiveness of two-stage interactive prompting. Instead of using two turns *cross-lingual alignment prompting* and *task-specific solver prompting* to separately perform alignment and task solving, we directly concatenate the *cross-lingual alignment prompting* and *task-specific solver prompting* using the newline character "\n" for LLM.

Results are illustrated in Figure 3. Compared with two-stage interactive prompting (CLP), we observe a significant average decrease of 10.4% in the single-turn prompting performance. We suppose that two-stage interactive prompts can better elicit the strong dialogue interactive ability of LLM, thereby enhancing the performance.

### 5.3.3 CLP is not a vanilla translation

As shown in Table 1, we can find that CLP even achieves a 2.2% higher average accuracy than `Translate-En`, which indicates that CLP is not a vanilla translation but utilizes the semantic alignment between the languages. To further understand how CLP works better than translation, we randomly choose 200 samples from different lan-

| Model | ET | HT | ID | IT | QU | SW | TA | TH | TR | VI | ZH | AVG |
|---|---|---|---|---|---|---|---|---|---|---|---|---|
| mT0-XXL (Muennighoff et al., 2022) | | | | | | | | | | | | |
| En-CoT | 24.2 | 23.2 | 5.2 | 23.0 | **29.4** | 7.4 | 31.0 | 16.6 | 29.2 | 34.8 | 10.2 | 21.3 |
| CLP | **41.4** | **30.8** | **20.6** | **30.8** | 21.6 | **34.4** | **33.6** | **33.6** | **32.6** | **49.2** | **12.2** | **32.1** |
| Bloomz-7B (Muennighoff et al., 2022) | | | | | | | | | | | | |
| En-CoT | 21.8 | 24.2 | 50.6 | 41.6 | 41.4 | **48.6** | 53.8 | 38.4 | 37.6 | **47.0** | **64.2** | 42.7 |
| CLP | **49.0** | **49.6** | **58.0** | **48.8** | **50.6** | 47.6 | **57.8** | **52.0** | **50.2** | 45.2 | 54.2 | **51.2** |
| llama-2-13B (Touvron et al., 2023) | | | | | | | | | | | | |
| En-CoT | 39.6 | 32.5 | 58.4 | 55.8 | **47.2** | 34.6 | 47.4 | 33.2 | 43.0 | **59.6** | 50.4 | 45.6 |
| CLP | **44.8** | **48.2** | **64.4** | **70.2** | 46.6 | **47.0** | **47.8** | **46.4** | **51.2** | 58.4 | **51.4** | **52.4** |

Table 2: The Acc. comparison on smaller and open-resource LLMs.

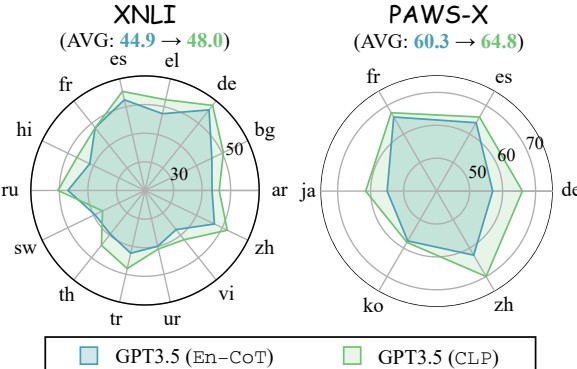

Figure 5: The Acc. comparison on other benchmarks.

guages for fine-grained exploration.

First, we find that CLP has 7 different strategies (as shown in Table 7), which all contribute to the final performance, which demonstrates the effectiveness of CLP. Further, we find that breaking down stage 1 further can help improve. Breaking down the actions of stage 1 into 2 to 4 strategies can significantly enhance performance (by at least 6.5%). For example, By decomposing the alignment process into "Problem Restatement" and "Preliminary Solution", better performance can be achieved, reaching 64.7% (an increase of 11.8% compared with Native-CoT).

### 5.3.4 How does prompt selection affect CLP?

We validate the robustness of the zero-shot cross-lingual chain-of-thought against the cross-lingual alignment prompts.

Table 4 illustrates the performance of 4 different cross-lingual alignment prompts. The experimental results demonstrate that although there are some fluctuations in the AVG Acc. of alignment and reasoning based on specific prompts (with a maximum difference of over 4%), all cross-lingual alignment prompts can still improve the performance compared to the traditional CoT. This further verifies the effectiveness of CLP.

### 5.3.5 Generality Analysis of CLP

In order to further study the generality of our work, we verify the generality of CLP from two aspects:

**CLP works well on other benchmarks.** We conduct experiments on other multilingual reasoning datasets, namely XNLI (Conneau et al., 2018) and PAWS-X (Yang et al., 2019). As shown in Figure 5, CLP can obtain better performance across a majority of languages. In comparison to En-CoT, we observed an average improvement of 3.1% on XNLI and 4.5% on PAWS-X[2].

**CLP works well on other LLMs.** To better understand the model generalization, we conduct the experiments on the XCOPA with smaller LLMs. Experimental results (as shown in Table 2) demonstrate that on smaller LLMs, CLP achieves at least a 6.8% improvement compared to En-CoT. Those further demonstrate the effectiveness and the wide applicability of CLP.

### 5.3.6 CLP can be further improved by in-context-learning

In recent years, in-context-learning (ICL) has achieved amazing results on LLMs. In order to further explore the performances of CLP within the ICL framework, a series of experiments were conducted. Subsequent analysis of the empirical findings has led to the following observations:

**Using ICL in cross-lingual alignment prompts can significantly enhance reasoning performance.** As depicted in Table 5, CLP exhibits a noteworthy 6.9% improvement over the zero-shot setting on MGSM. This further underscores the versatility of our approach as a plug-and-play solution, orthogonal to ICL methods, mutually reinforcing each other to augment performance.

---

[2]Due to the cost constraint, we randomly select 200 samples per language from test set.

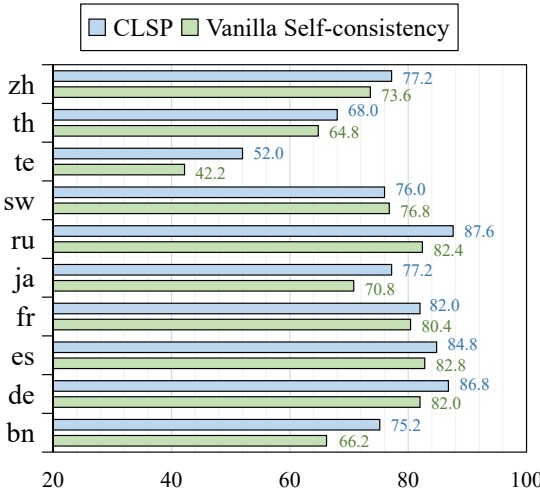

Figure 6: The accuracy comparison between Cross-lingual Self-consistent Prompting and Vanilla Self-consistency on MGSM benchmark.

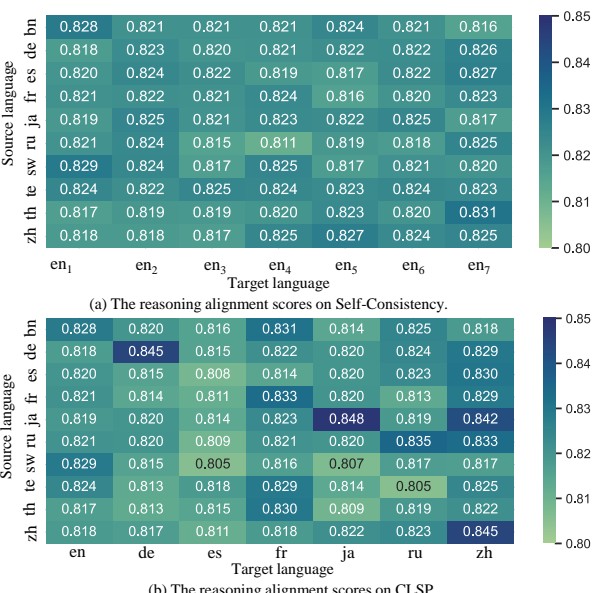

(a) The reasoning alignment scores on Self-Consistency.

(b) The reasoning alignment scores on CLSP.

Figure 7: The reasoning alignment score comparison between Cross-lingual Self-consistent Prompting and Vanilla Self-consistency.

**Using ICL in task-specific solver prompting can further boost reasoning performance.** As depicted in Table 5, the results reveal an additional 1.1% performance enhancement when incorporating Complex-CoT (Fu et al., 2023) in task-specific solver prompting. This further solidifies the distinctiveness of our approach in contrast to other CoT optimization methods, underscoring its adaptability and its capacity to offer more extensive support to downstream CoT inference techniques.

**For alignment, the example selection plays a pivotal role.** We conducted experiments with various combinations of Few-shot strategies. As shown in Table 6, if few-shot relies on a single strategy, the model's average performance drops significantly to 63.5%, even far below the effect of zero-shot. Conversely, when a more diverse set of strategies is employed within Few-shot examples, the model's performance shows a substantial improvement, reaching 75.9%. It shows that more diverse strategy samples lead to better performance enhancement.

### 5.4 CLSP Analysis

#### 5.4.1 Cross-lingual self-consistent prompting surpasses vanilla self-consistency

To validate the effectiveness of CLSP, we conduct experiments on vanilla self-consistency (VSC) (Wang et al., 2022) which obtains diverse CoT paths for better results. As shown in Figure 6, CLSP outperforms VSC about 4.5% on average, which verifies the effectiveness of CLSP. Further, we try to explore why CLSP works. We evaluate

the alignment scores between cross-lingual CoT inference paths (including CLSP and VSC) with all correct predicted results and manually annotated CoT inference paths. As illustrated in Figure 7, the variance of alignment scores generated by CLSP is significantly higher than VSC compared with the results of Yu et al. (2023). It shows that CLSP better ensembles language knowledge to enhance the final cross-lingual CoT performance. The implementation details are shown in Appendix A.3.1.

#### 5.4.2 More languages can not bring more improvement

A natural question that arises is, "*Does integrating a larger number of languages in self-consistent cross-lingual prompting lead to better overall performance?*" To answer this question, we explore the relationship between performance and the number of languages integrated. Some studies (Blevins and Zettlemoyer, 2022; Malkin et al., 2022) suggest that the LLM's performance is highly related with the proportion of pretraining data in each language. Therefore, we examine the language distribution (refer to Figure 8) in the widely used multilingual pretraining dataset, Common Crawl 2021. Based on the proportions, we incrementally integrated languages in descending and ascending order of their respective proportions. The results in Figure 9 demonstrate that in high-resource settings (>4%), performance improves as more languages are added. However, when incorporating low-

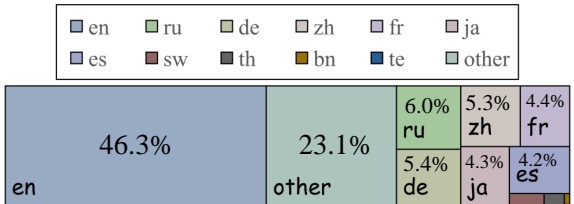

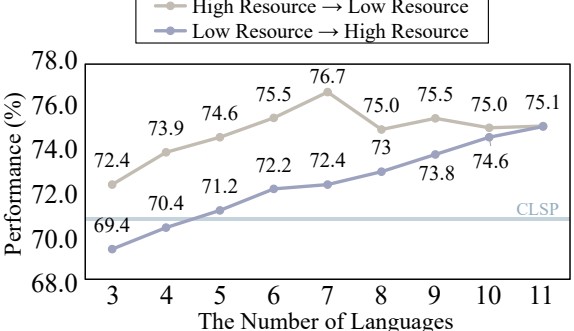

Figure 8: The language distribution of Common Crawl in 2021.

Figure 9: The impact of integrating languages on the final performance. Different colors represent different integration language sequences.

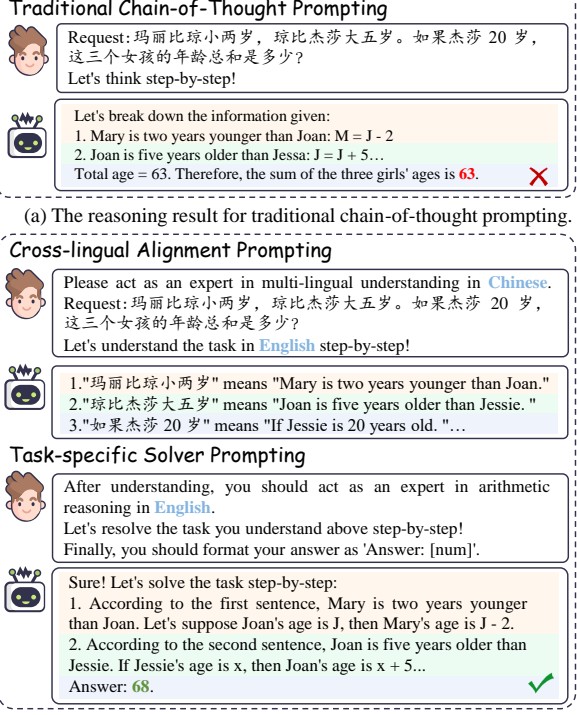

(a) The reasoning result for traditional chain-of-thought prompting.

(b) The reasoning result for cross-lingual chain-of-thought prompting.

Figure 10: Case Study.

resource languages, performance decreases with an increasing number of languages. These findings emphasize that the effectiveness of language integration is not solely determined by the number of languages integrated. Quantity of pretraining data for each language, especially in high-resource languages, play a crucial role. Balancing multiple languages considering available resources and impact is vital. This research provides valuable insights for developing multilingual models that strike a balance between incorporating diverse languages and maintaining high-performance standards.

### 5.4.3 Qualitative analysis

To further understand why CLP works intuitively, we provide a case study that compares the outputs generated by the traditional CoT approach and CLP. As depicted in Figure 10, we observe that the traditional CoT fails to comprehend all the information present in the query (missing the information about "Jessie is 20 years old"), thereby resulting in the error inference of the final result. However, our proposed CLP overcomes this limitation by first utilizing the cross-lingual alignment prompting to ensure the model comprehensively understands the given query, which detailed aligns the source language to the target language sentence-by-sentence. Then the task-specific solver prompting is implied to solve this problem step-by-step without deviation from

the information in the query. This indicates that our proposed CLP can simulate the model's ability to understand the cross-lingual query clearly before attempting to solve the problem. And this capability is essential because if the misunderstood happened, the final result may also be erroneously inferred in a high probability. This observation further validates the effectiveness of CLP.

### 5.4.4 Extension to XCOPA

To further verify the effectiveness of CLSP, we conduct experiments on XCOPA (Ponti et al., 2020), a widely adopted dataset for assessing commonsense reasoning skills across 11 different languages.

As the results presented in Table 3, in comparison to the baselines, we observe a significant average improvement of 4.7% in CLP performance. And it even surpasses the results reasoning with translated requests by 1.8%. Furthermore, CLSP leads to an additional enhancement of 7.4% compared to CLP. These results signify that apart from excelling in mathematical reasoning, both CLP and CLSP demonstrate notable effectiveness in addressing common-sense reasoning tasks.

## 6  Related Work

**Chain-of-Thought (CoT)** (Wei et al., 2022; Kojima et al., 2022) is an effective and step-by-step

| Model | ET | HT | ID | IT | QU | SW | TA | TH | TR | VI | ZH | AVG |
|---|---|---|---|---|---|---|---|---|---|---|---|---|
| GPT-3 (text-davinci-002) | | | | | | | | | | | | |
| `Direct` (Shi et al., 2022) | 73.8 | 55.6 | 88.8 | 95.4 | 51.2 | 56.0 | 54.6 | 70.2 | 88.6 | 80.4 | 91.4 | 73.3 |
| `En-CoT`† (Shi et al., 2022) | 88.8 | 79.6 | 91.4 | 96.6 | 52.2 | 67.4 | 55.8 | 84.2 | 91.2 | 86.6 | 93.4 | 80.7 |
| GPT-3.5 (gpt-3.5-turbo) | | | | | | | | | | | | |
| `Direct` (Ahuja et al., 2023) | 90.6 | 72.0 | 90.4 | 95.2 | 54.6 | 82.0 | 59.0 | 77.6 | 91.0 | 83.6* | 90.4* | 80.6 |
| `Translate-En` (Ahuja et al., 2023) | 88.2 | 79.4 | 90.8 | 94.4 | 50.0 | 77.6 | **87.0** | 82.2 | 87.8 | 88.4* | 92.2* | 83.5 |
| CLP | 89.6 | 79.4 | 94.2 | 92.8 | 63.6 | 84.8 | 73.4 | 87.8 | 91.2 | 90.8 | 91.2 | 85.3 |
| CLSP | **96.8** | **90.6** | **95.2** | **95.8** | **85.8** | **92.8** | 83.2 | **93.2** | **96.8** | **94.2** | **95.8** | **92.7** |
| HUMAN (Ponti et al., 2020) | 98.2 | 96.4 | 100.0 | 97.0 | 94.8 | 99.0 | 98.6 | 98.2 | 96.4 | 98.4 | 96.6 | 97.6 |

Table 3: Main results of XCOPA, * denotes the reproduction result. † denotes the 6-shot results.

strategy applied to Large Language Models (LLMs) for zero-shot and few-shot reasoning. CoT prompts, which can be a single instruction or a set of CoT examples, facilitate the generation of intermediate reasoning steps. Recently, a series of studies (Zhou et al., 2022; Wang et al., 2022, 2023c; Khot et al., 2023) have proposed their respective prompting strategies, dividing the entire task into smaller sub-tasks and subsequently resolving, planning, and executing these subtasks. With the improvement in model capabilities, some works (Zelikman et al., 2022; Zhou et al., 2023; Hu et al., 2023; Gao et al., 2023) treat instructions as "programs" for further search, execution, or optimization. Building upon this, considering the feedback brought by execution, ReAct (Yao et al., 2023) and Reflexion (Shinn et al., 2023) further explore the interactive generation of inference decisions and task execution, thereby achieving greater synergy.

**Cross-lingual Generalization** Prior studies have demonstrated the benefits of pre-trained multilingual models in diverse downstream tasks, such as cross-lingual spoken language understanding (Qin et al., 2020, 2022; Zheng et al., 2022) and cross-lingual summarization (Wang et al., 2023a,b; Bhattacharjee et al., 2023). Recently, with the emergence of Large Language Models (LLMs), non-training-based cross-lingual learning has gained more attention (Brown et al., 2020; Ahuja et al., 2023; Winata et al., 2023; Zeng et al., 2023; Huang et al., 2023). Additionally, in the context of cross-lingual alignment, the current common practice involves employing few-shot learning to guide models for better alignment (Winata et al., 2021; Shi et al., 2022; Tanwar et al., 2023; Lin et al., 2022).

Compared to their work, we explore the zero-shot cross-lingual alignment CoT and introduce CLP to address this problem, which does not need any additional examples to be constructed. Furthermore, we explore Cross-lingual Self-consistent Prompting (CLSP) to enhance the performance by leveraging chained cross-lingual pathways devised by experts in various languages.

## 7 Conclusion

In this work, we introduced cross-lingual prompting (CLP) for cross-lingual Chain-of-Thought. Specifically, CLP consists of *cross-lingual alignment prompting* and *task-specific solver prompting* to align representations across languages and generate the final reasoning paths in cross-lingual settings. In addition, we proposed a cross-lingual self-consistent prompting (CLSP) to effectively leverage knowledge across languages, which further boosts performance over CLP. Extensive experiments reveal that both CLP and CLSP can attain promising performance in cross-lingual CoT.

## Acknowledgements

This work was supported by the National NaturalScience Foundation of China (NSFC) via grant 62306342, 62236004 and 61976072. This work was also sponsored by CCF-Baidu Open Fund. We are grateful for resources from the High Performance Computing Center of Central South University. Libo Qin is the corresponding author.

## Limitations

Consistent with the findings of Kojima et al. (2022); Zhu et al. (2023), our results also indicate that CLP exhibits varying performance improvements in reasoning based on different prompts. Although all of these prompts can enhance the performance, there are still significant performance disparities, with differences exceeding 4%. Therefore, enhancing the robustness of model alignment remains an urgent issue that needs to be addressed in the future.

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

## A  Appendix

### A.1  Robust Analysis Implementation

In order to further verify the robustness of CLP, we conducted an analysis of the final results for various CLPs with different expressions. Specifically, we utilize GPT3.5 to generate 3 guiding prompts synonymous with "Let's understand the task in English step by step!". Our instruction is as follows:

> Assuming you are a professional rewriter, you need to modify the following request into three different versions:.
> Let's think in [Target Language $L_t$] step by step!

The final generated prompt and corresponding results are shown in Table 4.

### A.2  Chain-of-Thought Quality Scoring Implementation

The ROSCOE framework (Golovneva et al., 2022) incorporates multiple chain-of-thought quality metrics, with the reasoning alignment vector $\alpha = r\text{-align}(h \rightarrow s) = \{\alpha_1, \alpha_2, \cdots, \alpha_N\} \in [0,1]^N$ from the $N$-step hypothesis $h = \{h_i\}_{i=1}^N$ to the source input $s$ of length $T$, where $\alpha_i$ are defined as:

$$r\text{-align}(h_i \rightarrow s) = \frac{1 + \max_{j=1}^T \cos(h_i, s_j)}{2}. \quad (5)$$

#### A.2.1  Faithfulness

The Faithfulness ($F$) score is calculated based on the alignment between the hypothesis steps $h$ and the source sentences $s$. It represents the average reasoning alignment score over the steps of reasoning:

$$F = \frac{1}{N} \sum_{i=1}^N r\text{-align}(h_i \rightarrow s). \quad (6)$$

The Faithfulness score serves as a measure to assess whether the model misconstrued the problem statement or if the reasoning chain is characterized by vagueness, irrelevance, or the misuse of information.

#### A.2.2  Informativeness Step

Informativeness-Step (Info-Step) measures the utilization of information from the source text $s$ in the reasoning steps $h$:

$$\text{Info-Step} = \frac{1}{2T} \sum_{t=1}^T r\text{-align}(s_t \rightarrow h) + \frac{1}{2}F. \quad (7)$$

Info-Step assigns a higher score to reasoning steps that demonstrate a strong alignment with the source, thereby indicating the extent to which the generated hypothesis incorporates the information from the source. Conversely, a lower Info-Step score indicates reasoning steps that are unrelated to the source sentences or overlook the provided information in the context.

#### A.2.3  Informativeness Chain

Just like the Info-Step metric, the Informativeness-Chain (Info-Chain) metric measures the extent of concordance between the hypothesis chain and the source. The calculation is as follows:

$$\text{Info-Chain} = \frac{1 + \cos(h, s)}{2}. \quad (8)$$

To facilitate this computation, we treat the reasoning chain and the source context as an integrated entity.

#### A.2.4  Missing Step

To pinpoint any significant steps that could be lacking in the hypothesis, (Golovneva et al., 2022) introduce the Missing Step (Miss-Step) metric, which examines the alignment between the reference reasoning text $r = \{r_i\}_{i=1}^K$ and the hypothesis $h$. Miss-Step is needed to meticulously assess each step in the reference and verify the existence of a similar step in the hypothesis. The metric is computed as:

$$\text{Miss-Step} = \min_{i=1}^K (r\text{-align}(r_i \rightarrow h)). \quad (9)$$

#### A.2.5  Multi-lingual Setting

Due to the limited support of the original ROSCOE (Golovneva et al., 2022) framework for monolingual English, we expanded ROSCOE to operate in a cross-lingual setting to enhance the assessment of Cross-lingual CoT's inference quality. For the backbone of sentence similarity computation in the model, we employed a multilingual variant of MP-Net[3] (Reimers and Gurevych, 2019).

### A.3  Reasoning Alignment Scoring

#### A.3.1  Metric Definition

Reasoning Alignment Scoring (RAS) offers a simple method to evaluate the accuracy of the hypothesis chain by examining the extent of overlap between the hypothesis and the reference. One ap-

---

[3]https://huggingface.co/sentence-transformers/paraphrase-multilingual-mpnet-base-v2

| Cross-Lingual Prompting | bn | de | es | fr | ja | ru | sw | te | th | zh | AVG |
|---|---|---|---|---|---|---|---|---|---|---|---|
| ● Let's understand the task in English step by step! | **64.8** | **80.0** | 81.6 | 79.2 | **69.2** | **82.4** | **74.8** | 38.8 | **62.0** | 73.6 | **70.6** |
| ● We should grasp the task in English by breaking it down into steps. | 50.8 | 69.6 | **84.8** | **82.4** | 77.6 | 80.8 | 73.2 | 36.0 | 61.6 | 73.6 | 69.0 |
| ● Step by step, let's comprehend the task in English! | 60.8 | 76.8 | 82.4 | 76.0 | 70.0 | 78.8 | 67.6 | **40.8** | 57.6 | 72.4 | 68.3 |
| ● Our approach should involve understanding the task in English through a step-by-step process. | 56.4 | 80.8 | 77.6 | 81.6 | 66.4 | 72.8 | 66.4 | 35.6 | 63.2 | 64.0 | 66.5 |
| GPT3.5 (`En-CoT`) | 50.0 | 73.6 | 69.6 | 70.0 | 60.4 | 65.6 | 55.2 | 22.0 | 48.0 | 63.2 | 57.8 |

Table 4: Performance of different prompts in CLP.

| Model | bn | de | es | fr | ja | ru | sw | te | th | zh | AVG |
|---|---|---|---|---|---|---|---|---|---|---|---|
| CLP | 65.0 | 80.0 | 82.0 | 79.0 | 63.0 | 84.0 | 63.0 | 44.0 | 60.0 | 70.0 | 69.0 |
| CLP(3-shot) | 76.0 | 85.0 | 84.0 | 75.0 | 80.0 | 87.0 | 68.0 | 61.0 | 59.0 | 84.0 | 75.9 |
| CLP(3-shot) +Complex-CoT (Fu et al., 2023) | 71.0 | 89.0 | 85.0 | 81.0 | 86.0 | 88.0 | 72.0 | 50.0 | 61.0 | 87.0 | 77.0 |

Table 5: Additional Experiment on Few-shot Setting.

proach to achieving this is by quantifying the reasoning alignment between the two, which can be calculated as:

$$RAS = \frac{1}{N}\sum_{i=1}^{N} r\text{-align}(h_i \rightarrow r). \quad (10)$$

### A.3.2 Implementation Setting

Since completely incorrect reasoning can also lead to a significant decrease in RAS, we conducted the experiments in Figure 7 by excluding all samples with prediction errors and only calculating RAS on correctly predicted samples.

In Figure 7 (a), we selected English as the target language and generated seven CoT reasoning results by adjusting the model's output temperature. We calculated the RAS between the reasoning step outputs of each correctly predicted sample and the standard reasoning step outputs. By averaging the RAS of all samples, we obtained the comprehensive RAS for source-to-English comprehension. Similarly, in Figure 7 (b), we chose a high-resource language as the target language and obtained seven CoT reasoning results. We computed the RAS between the reasoning step outputs of each correctly predicted sample and the standard reasoning step outputs, and then averaged the RAS of all samples to obtain the comprehensive RAS for source-to-target language comprehension.

Overall, the CLSP exhibits a stronger diversity in reasoning paths, particularly in the original language reasoning of zh, ja, and de, which shows a higher similarity to the original reasoning paths ($\geq 0.845$). On the other hand, cross-lingual reasoning from es to sw, ja to sw, and ru to te demonstrates more unique reasoning paths ($\leq 0.805$).

### A.4 Strategy Definition

In our deep exploration, we find that CLP not only serves as simple translation but also has seven different strategies, which are summarized below:

- **Step-by-step Translation**: The model divides the translation process into steps based on commas or periods and translates them step by step, as illustrated in Figure 10.

- **Key Information Extraction**: The model first extracts key terms or critical conditions from the request for translation. This aids the model in achieving better cross-lingual alignment.

- **Preliminary Solution**: This strategy indicates that CLP starts preliminary mathematical operations based on comprehension. It may even provide answers during the alignment phase. However, the model's second stage may modify this answer, so it is not the final solution.

- **Complete translation**: This strategy indicates that the model directly performs machine translation of the request without sentence splitting or step-wise operations.

- **Problem restatement**: This strategy indicates that the model rephrases the request. Unlike machine translation, problem restatement requires the model to infer, add its understanding, and include information inferred from the request through reasoning.

| # Strategy | bn | de | es | fr | ja | ru | sw | te | th | zh | AVG |
|---|---|---|---|---|---|---|---|---|---|---|---|
| 3 | 76.0 | 85.0 | 84.0 | 75.0 | 80.0 | 87.0 | 68.0 | 61.0 | 59.0 | 84.0 | 75.9 |
| 2 | 65.0 | 85.0 | 82.0 | 73.0 | 66.0 | 80.0 | 69.0 | 46.0 | 57.0 | 68.0 | 69.1 |
| 1 | 57.0 | 78.0 | 73.0 | 73.0 | 59.0 | 74.0 | 57.0 | 40.0 | 59.0 | 65.0 | 63.5 |

Table 6: Performance of different number of strategies. Strategies are adopted from large to small according to the strategy proportion in Table 7.

| Strategy | CLP Acc. | Native-CoT Acc. | Ratio (%) |
|---|---|---|---|
| Step-by-step Translation | 61.11 | 38.89 | 9.00 |
| Key Information Extraction | 60.00 | 60.00 | 5.00 |
| Preliminary Solution | 61.11 | 54.63 | 54.00 |
| Complete Translation | 58.33 | 50.00 | 24.00 |
| Problem Restatement | 57.28 | 50.49 | 51.50 |
| Step Division | 65.96 | 51.06 | 23.50 |
| Code-Switching | 62.50 | 62.50 | 4.00 |
| Denial of Service | 50.00 | 42.86 | 7.00 |

Table 7: The effectiveness and distribution of the different strategies. There are more details in Appendix A.4.

Furthermore, in order to explore the impact of different examples on CLP, we further analyze the impact that examples using different alignment strategies mentioned in Section 5.3.3 can have on downstream tasks. We manually annotate the dev set and used multiple strategies for annotation. Experiments in Table 6 show that as the diversity of strategies increases, the performance of the model gradually increases.

- **Step Division**: This strategy encompasses two situations: (1) The model actively divides the cross-lingual alignment process into multiple steps. For example, it will divide the alignment process into "Step 1: Identify the context and topic", "Step 2: Translate the sentence" and "Step 3: Analyze the sentence structure". (2) The model actively plans the next task and divides the request into several sub-questions.

- **Code-switching**: This strategy indicates that the model actively replaces certain words in the text with words from the target language.

- **Denial of Service**: ChatGPT refuses to perform cross-lingual alignment and delegates alignment directly to the second stage.

## A.5 Few-shot Setting

In order to verify the effect of CLP on ICL, we further designed experiments with few-shot settings for analysis. Specifically, we first selected 1,000 samples of data from MGSM test set for testing. In the alignment stage, we immediately used some examples from the dev set to construct cross-language alignment examples. The results of these examples were all generated by GPT3.5. We only keep the correct answers as examples.

In the problem-solving phase, we further used the example of Complex-CoT as a problem-solving example. The results in Table 5 show that the two-stage ICL can better promote the performance of the model. This also illustrates the versatility of CLP and its ability to be orthogonal to other prompt optimization solutions.