# OpenReview forum: "Cross-lingual Prompting: Improving Zero-shot Chain-of-Thought Reasoning across Languages"
_EMNLP/2023/Conference — EMNLP 2023 Main_

### Official Review · Reviewer_wcFZ · 2023-08-01

**Typos Grammar Style And Presentation Improvements:** Line 289
**Soundness:** 3

**Excitement:**

3: Ambivalent: It has merits (e.g., it reports state-of-the-art results, the idea is nice), but there are key weaknesses (e.g., it describes incremental work), and it can significantly benefit from another round of revision. However, I won't object to accepting it if my co-reviewers champion it.

**Missing References:**

Fu et al., Complexity-Based Prompting for Multi-step Reasoning. ICLR 2023

**Paper Topic And Main Contributions:**

The authors tackle the problem of Chain of Thought (CoT) Reasoning in a multilingual setting. Specifically, the authors propose extending CoT to multiple languages via 2 major sequential prompting components (1) Cross-lingual Alignment Prompting and (2) Task-specific Solver Prompting. While the former focuses on aligning representations across different languages, the latter's goal is to generate CoT outcomes for the reasoning task.

**Questions For The Authors:**

A. Table 1: Why are EN-CoT, Native-CoT missing for GPT3.5 backbone?

B. What is the difference between "Cross-lingual Alignment" and "Translating each step of the request from source language to English"? The sample in Figure 2,9 creates the sense that the LLMs are prompted to simply conduct translation rather than understanding of the request. What is the multilingual understanding here?

C. Figure 6: The difference between demonstrated alignment scores on CLSP and SC does not seem significant. What is the significance mentioned in Line 354-358? In addition, as the calculation of the score is dependent on the correctly predicted samples, will the calculation result in incomparable scores among evaluated languages?

D. How does leveraging Complex-CoT [1] instead of SC-CoT affect the performance?

E. Figure 8: One might argue the drop in performance when adding later languages might be due to the capacity of LLMs, leading to forgetting/ slower adaptation of LLMs. How is the performance change  when adding the languages in the reverse order (i.e. low-resource to high-resource languages)? This addition helps verify the claim "adding low-resource languages decreases the performance with increasing number of languages" (Line 382-384).

[1] Fu et al., Complexity-Based Prompting for Multi-step Reasoning. ICLR 2023

**Reasons To Accept:**

1. The paper is well-written. The proposed method is intuitive and easy to follow.

2. Extensive empirical studies are conducted.

3. Promising improvements over the baselines are quite significant across different languages.

**Reasons To Reject:**

1. The claim that the proposed method is plug-and-play is not sufficiently substantiated. Although 3 backbone models are presented (GPT-3, PaLM-540B and GPT3.5), the empirical study is only based on GPT3.5

2. The method misses an important ablation study of comparing the proposed method with  GPT3.5-Translate-EN: Non-English input request is translated to English via GPT3.5 Machine Translation Engine (Table 1). It is possible the improvements are mostly done via the quality of MT from GPT3.5.

3. Some claims are not fully substantiated by the empirical studies (See Questions).

**Reproducibility:**

3: Could reproduce the results with some difficulty. The settings of parameters are underspecified or subjectively determined; the training/evaluation data are not widely available.

**Reviewer Confidence:**

4: Quite sure. I tried to check the important points carefully. It's unlikely, though conceivable, that I missed something that should affect my ratings.

---

> ### Author Rebuttal · Authors · 2023-08-28
>
> Thanks for your thoughtful and comprehensive feedback on our work. We sincerely appreciate your valuable insights and we will address concerns point to point.
>
> **Qustion1**: Table 1: Why are EN-CoT, Native-CoT missing for GPT3.5 backbone?
>
> **Answer1**: Thanks for your kind mention. Actually, we had included the EN-COT and Native CoT result for GPT3.5, which is illustrated in Table 1(in the fourth line from the bottom) and Figure 10 (in Appendix). For a more fair comparison, our Native-CoT also adopts a two round setting, with the only difference from CLP being the lack of cross language alignment. This is different from the single round settings of other baselines, so it is not placed in the main table. We will follow your suggestion and add Native-CoT to Table 1 in the next version to make the work more clearer.
>
> **Question2**: Difference between "Cross-lingual Alignment" and "Translating each step of the request from source language to English"?
>
> **Answer2**: Thanks for your insightful comment. We sincerely think that Cross-lingual Alignment is not solely a simple translation. As shown in Figure 2, Cross-lingual alignment proactively divides each step (玛丽比琼小两岁, 琼比杰莎大五岁, 如果杰莎 20 岁…) and progressively translates them ("Mary is two years younger than Joan.", "Joan is five years older than Jessie. "). The behind intuition of Cross-lingual Alignment is that when the model can gradually understand the problem correctly across languages, it can have better cross-language alignment ability. We will add more discussion in the next version.
>
> **Question3**: Figure 6: The difference between demonstrated alignment scores on CLSP and SC does not seem significant.
>
> **Answer3**:Thanks for your constructive feedback. Due to the fact that most of the gold paths for solving mathematical problems are consistent, most of the predicted correct CoT inference paths are also consistent, which leads to very high alignment scores. Therefore, the alignment scores from 0.80 to 0.84 in the figure are significant to this certain extent, which is consistent with the observation of [1].
>
> [1] Yu et al., ALERT: Adapt Language Models to Reasoning Tasks. ACL2023
>
> **Question4**: How does leveraging Complex-CoT [1] instead of SC-CoT affect the performance?
>
> **Answer4**: Thanks for your insightful comment. Complex CoT is an promising approach using a series of hard and multi-step reasoning ICL examples, which can greatly enhance the model's complex logical reasoning ability. Our work is a plug-and-play module that can be completely orthogonal to Complex CoT. We will follow your suggestion to explore the role of Complex CoT in our framework. Specifically, we will replace the simple “Let’s resolve the task step by step” trigger to the hard and multi-step examples selected from [1] for better reasoning.
>
> [1] Fu et al., Complexity-Based Prompting for Multi-step Reasoning. ICLR 2023
>
> **Question5**:How is the performance change when adding the languages in the reverse order (i.e. low-resource to high-resource languages)?
>
> **Answer5**: Thank you for your thoughtful suggestion. We have performed experiments by integrating non-English languages in reverse order, from low to high resources. We found two obvious phenomena.  First, the performance of the integration has dropped significantly, and the performance is even lower than that of pure English CLSP, which shows that the integration of low-resource languages is harmful to the performance improvement of the model. Second, we noticed that the integration performance from low resource to high resource increases gradually, but the integration performance from high resource to low resource decreases finally, which further verifies our claim that the integration of high-resource languages is more critical to the performance improvement of CLSP. We will add more analysis in the next version.
>
> **We would greatly appreciate it if you can re-consider the work considering our clarification.**

---

### Official Review · Reviewer_prAk · 2023-08-04

**Typos Grammar Style And Presentation Improvements:** 1) Line 52
**Soundness:** 4

**Excitement:**

3: Ambivalent: It has merits (e.g., it reports state-of-the-art results, the idea is nice), but there are key weaknesses (e.g., it describes incremental work), and it can significantly benefit from another round of revision. However, I won't object to accepting it if my co-reviewers champion it.

**Missing References:**

Zero-Shot CoT Promptings

Cross-lingual Transfer Learning.

**Paper Topic And Main Contributions:**

The paper proposes a novel approach called Cross-lingual Prompting (CLP) to generalize the success of zero-shot chain-of-thought (CoT) reasoning to different languages. The paper presents two components of their approach: (1) Cross-lingual Alignment Prompting, which aligns representations between different languages using an English understanding prompt, and (2) Task-specific Solver Prompting, which completes the final task by using a resolver prompt. Additionally, this paper proposes Cross-lingual Self-consistent Prompting (CLSP) to ensemble reasoning paths across languages. The experiments are conducted on the MGSM and XCOPA datasets, and the results demonstrate that CLP achieves state-of-the-art performance with an average gain of over 5 points compared to all baselines.

**Questions For The Authors:**

1) The proposed method employs alignment prompting to obtain understanding in English. The aim is to compare this method with the baseline approach of first obtaining an understanding of the source language and then translating it into the target language.

2) According to Table 2, the results indicate that performance is influenced by the specific wording used in prompts. It is important to know how to find the optimal wording to use.

3) How well does the proposed prompting perform in the few-shot setting?

4) How well does the proposed prompting perform with smaller LLMs like mT5, BLOOM, GPT-Neo, LLaMa, and ChatGLM?

I will update the final score according to the authors responses.

**Reasons To Accept:**

1) The proposed Cross-lingual Self-consistent Prompting approach is intuitive, simple, and effective.

2) The experiments conducted on multiple datasets and the comparison with various baselines demonstrate the effectiveness of the proposed approach.

**Reasons To Reject:**

1) The literature review needs improvement. Numerous papers on zero-shot CoT and cross-lingual transfer learning have been published this year. The paper should incorporate more related works on zero-shot CoT prompting and cross-lingual transfer learning.

2) This paper should verify the proposed prompting on more cross-lingual datasets mentioned by previous work, such as MKQA, XNLI, and XL-WiC.

**Reproducibility:**

5: Could easily reproduce the results.

**Reviewer Confidence:**

4: Quite sure. I tried to check the important points carefully. It's unlikely, though conceivable, that I missed something that should affect my ratings.

---

> ### Author Rebuttal · Authors · 2023-08-28
>
> Thanks for your acknowledgements and interests on our work! We sincerely appreciate your thorough comment of our work and we will address each of your concerns below:
>
> **Question1**: literature review needs improvement
>
> **Answer1**: Thanks for your kind mention. We will follow your suggestion to add more related literature work in the next version, including zero-shot CoT prompting and cross-lingual transfer learning.
>
> **Question2**: It is important to know how to find the optimal wording to use.
>
> **Answer2**: Thanks for your insightful comment. Yes, we totally agree with your perspective and this is an interesting research question to investigate. In our work, we drew inspiration from ChatGPT, a conversational LLM, and devised a multi-turn iterative prompting approach to enhance the multi-round interactive understanding capability of the LLM. Another design principle that I think to be quite important is considering task-aware prompts to better elicit the capabilities of large-scale models for solving specific task.
>
> **Question3**: How well does the proposed prompting perform in the few-shot setting?
>
> **Answer3**: Thanks for your constructive comment. Actually, we conducted the few-shot setting by providing some demonstrations in the preliminary experiments. The results demonstrate that adding demonstrations can bring improvement compared to the zero-shot setting, which is consistent with previous observations [1-3]. We will follow your suggestion to add more discussion in the next version.
>
> **Question4**: How well does the proposed prompting perform with smaller LLMs like mT5, BLOOM, GPT-Neo, LLaMa, and ChatGLM?
>
> **Answer4**: Thanks for your insightful comment. This is a very good question. Yes, we also consider the setting. When we apply mT0 in our work, we find that the results are close to 0, and there is no relevant cross-lingual CoT phenomenon observed, which is consistent with the previous observation [4,5]. We speculate that smaller LLM models might struggle with zero-shot cross-modal CoT scenarios. We will include more discussion in the next version, hoping to contribute some meaningful insights to this field.
>
> [1] Min et al., Rethinking the Role of Demonstrations: What Makes In-Context Learning Work? EMNLP2022.
>
> [2] Wei et al., Finetuned language models are zero-shot learners. ICLR2023.
>
> [3] Li et al., In-Context Learning with Many Demonstration Examples. arxiv2023.
>
> [4] Li et al., Symbolic Chain-of-Thought Distillation: Small Models Can Also “Think” Step-by-Step. ACL2023.
>
> [5] Xue et al., RCOT: Detecting and Rectifying FactualInconsistency in Reasoning by Reversing Chain-of-Thought. arxiv2023.

---

### Official Review · Reviewer_mrxR · 2023-08-07

**Soundness:** 4

**Excitement:**

4: Strong: This paper deepens the understanding of some phenomenon or lowers the barriers to an existing research direction.

**Paper Topic And Main Contributions:**

This paper introduces a new technique called cross-lingual prompting (CLP) that aims to improve zero-shot chain-of-thought reasoning across languages. By aligning representations across different languages and generating task-specific solver prompting, CLP can help improve reasoning accuracy in a wide range of tasks. the author also introduce  cross-lingual self-consistent prompting (CLSP) to ensemble different reasoning paths across languages. The experiments show that CLP and CLSP can significantly improve reasoning accuracy across languages, and that CLSP is particularly effective in improving reasoning accuracy for low-resource languages.

**Reasons To Accept:**

The paper studies an important problem -- multilingual zero-shot CoT prompting and presents a novel approach to address this problem. The authors introduce two main components of cross-lingual prompting (CLP): cross-lingual alignment prompting and task-specific solver prompting, as well as cross-lingual self-consistent prompting (CLSP) to ensemble different reasoning paths across languages. The experimental evaluations on the multilingual reasoning benchmark MGSM and XCOPA demonstrate that CLP and CLSP significantly outperform existing prompting methods and achieve state-of-the-art performance. The paper provides a clear and detailed description of the proposed approach and experimental results, and the findings have important implications for improving cross-lingual reasoning in a wide range of applications.

**Reasons To Reject:**

Overall I feel the paper is good -- clearly written and the proposed method is intuitive. A few places that can be further improved include:
1. More datasets on traditional multilingual tasks like XNLI, XTREME, to show the proposed technique can generalize to tasks with different levels of reasoning requirements.
2. Consider adding one experiment on open-source LLM as the current GPT-series and PaLM v1 is somewhat hard to reproduce entirely from the outsider.
3. Small typo around line 90 -- "Let’s resolver the task ..." to "Let's resolve the task"


**Reproducibility:**

3: Could reproduce the results with some difficulty. The settings of parameters are underspecified or subjectively determined; the training/evaluation data are not widely available.

**Reviewer Confidence:**

3: Pretty sure, but there's a chance I missed something. Although I have a good feel for this area in general, I did not carefully check the paper's details, e.g., the math, experimental design, or novelty.

---

> ### Author Rebuttal · Authors · 2023-08-28
>
> Thanks for your acknowledgements and interests on our work! We sincerely appreciate your thorough comment of our work.
>
> **Question1**: More datasets on traditional multilingual tasks like XNLI, XTREME
>
> **Answer1**: Thanks for your constructive suggestion. In this work, we follow the same setting as Shi et al [1] to consider the MGSM and XCOPA task. We acknowledge your suggestion regarding the XNLI, XTREME task to valide the different levels of reasoning requirements and follow your suggestion to add more discussion in the next version.
>
> **Question2**: adding some open-source LLMs for better reproducibility
>
> **Answer2**: Thanks for your insightful comment. We totally agree with your point and we will add some open-source LLMs in the next version to promote the community. In addition, all codes used in our work will be made public to facilitate the research.
>
> **Question3**: Small typo
>
> **Answer3**: Thanks for your kind mention. We will follow your suggestions to polish our work in the next version.
>
> [1] Shi et al., Language Models are Multilingual Chain-of-Thought Reasoners.

---

### Meta-Review · Area_Chair_atSq · 2023-09-20

**Recommendation:** 5

**Metareview:**

The paper proposes a novel approach called Cross-Lingual Prompting (CLP)  to improve zero-shot chain-of-thought reasoning across language. The experimental analysis show improvements over multilingual reasoning benchmarks MGSM and XCOPA.  The paper is well written and the approach and experiments are described well.

The draft can be improved by including more related literature work, adding discussion on few-shot prompting and including the analysis discussed in the rebuttal.

---

### Decision · Program_Chairs · 2023-10-07

**Decision:**

Accept-Main

**Comment:**

The paper proposes a novel approach called Cross-Lingual Prompting (CLP)  to improve zero-shot chain-of-thought reasoning across language. The experimental analysis show improvements over multilingual reasoning benchmarks MGSM and XCOPA.  The paper is well written and the approach and experiments are described well.

The draft can be improved by including more related literature work, adding discussion on few-shot prompting and including the analysis discussed in the rebuttal.